# Skeletal Muscle-Specific *Bis* Depletion Leads to Muscle Dysfunction and Early Death Accompanied by Impairment in Protein Quality Control

**DOI:** 10.3390/ijms24119635

**Published:** 2023-06-01

**Authors:** Soon-Young Jung, Tae-Ryong Riew, Hye Hyeon Yun, Ji Hee Lim, Ji-Won Hwang, Sung Won Jung, Hong Lim Kim, Jae-Seon Lee, Mun-Yong Lee, Jeong-Hwa Lee

**Affiliations:** 1Department of Biochemistry, College of Medicine, The Catholic University of Korea, Seoul 06591, Republic of Korea; syjjeong@hanmail.net (S.-Y.J.); nice1205@hanmail.net (H.H.Y.); 2Institute for Aging and Metabolic Diseases, College of Medicine, The Catholic University of Korea, Seoul 06591, Republic of Korea; 3Department of Anatomy, Catholic Neuroscience Institute, College of Medicine, The Catholic University of Korea, Seoul 06591, Republic of Korea; taeryong.riew@catholic.ac.kr (T.-R.R.); jidnjs2@naver.com (J.-W.H.); swjung@catholic.ac.kr (S.W.J.); munylee@catholic.ac.kr (M.-Y.L.); 4Division of Nephrology, Department of Internal Medicine, College of Medicine, The Catholic University of Korea, Seoul 06591, Republic of Korea; didsuai@hanmail.net; 5Integrative Research Support Center, Laboratory of Electron Microscope, College of Medicine, The Catholic University of Korea, Seoul 06591, Republic of Korea; wgwkim@catholic.ac.kr; 6Research Center for Controlling Intercellular Communication (RCIC), College of Medicine, Inha University, Incheon 22212, Republic of Korea; jaeslee@inha.ac.kr; 7Program in Biomedical Science & Engineering, Inha University, Incheon 22212, Republic of Korea

**Keywords:** Bcl-2-interacting cell death suppressor, skeletal muscle, HSP70, YAP1, filamin C

## Abstract

Bcl-2-interacting cell death suppressor (BIS), also called BAG3, plays a role in physiological functions such as anti-apoptosis, cell proliferation, autophagy, and senescence. Whole-body *Bis*-knockout (KO) mice exhibit early lethality accompanied by abnormalities in cardiac and skeletal muscles, suggesting the critical role of BIS in these muscles. In this study, we generated skeletal muscle-specific *Bis*-knockout (*Bis*-SMKO) mice for the first time. *Bis*-SMKO mice exhibit growth retardation, kyphosis, a lack of peripheral fat, and respiratory failure, ultimately leading to early death. Regenerating fibers and increased intensity in cleaved PARP1 immunostaining were observed in the diaphragm of *Bis*-SMKO mice, indicating considerable muscle degeneration. Through electron microscopy analysis, we observed myofibrillar disruption, degenerated mitochondria, and autophagic vacuoles in the *Bis*-SMKO diaphragm. Specifically, autophagy was impaired, and heat shock proteins (HSPs), such as HSPB5 and HSP70, and z-disk proteins, including filamin C and desmin, accumulated in *Bis*-SMKO skeletal muscles. We also found metabolic impairments, including decreased ATP levels and lactate dehydrogenase (LDH) and creatine kinase (CK) activities in the diaphragm of *Bis*-SMKO mice. Our findings highlight that BIS is critical for protein homeostasis and energy metabolism in skeletal muscles, suggesting that *Bis*-SMKO mice could be used as a therapeutic strategy for myopathies and to elucidate the molecular function of BIS in skeletal muscle physiology.

## 1. Introduction

Bcl-2 interacting cell death suppressor (BIS), also called BAG3, was originally identified as Bcl-2 and Hsp70/Hsc70 interacting protein [1,2]. Accumulating evidence from various types of cancer cells has revealed the anti-apoptotic function of BIS against various stimuli, such as cytokines, chemotherapeutic agents, and oxidative and proteotoxic stress [3,4]. In line with its pro-survival activity, BIS is highly expressed in various cancers, including breast, hepatocellular, lung, pancreatic, and thyroid carcinomas; lymphocytic leukemia; medulloblastoma; and melanoma, which is associated with poor patient prognosis [3,5,6]. In addition to its anti-apoptotic function, BIS was found to be involved in various cellular functions, such as regulation of cell proliferation, migration, senescence, autophagy, and protein homeostasis [3,7,8,9].

BIS is ubiquitously expressed in normal human tissues, particularly in skeletal muscles and the heart [1], indicating the important roles of BIS in these tissues. The essential functions of BIS have been confirmed by several findings. First, whole-body *Bis*-KO mice, which were independently established by our and other groups [10,11], commonly exhibited growth retardation, early lethality, and disintegration of z-disks in skeletal muscles [10,11]. These phenotypes suggest that BIS is crucial for survival, probably through the maintenance of normal physiological muscle function. Second, in recent decades, numerous mutations of BIS have frequently been observed in patients with myofibrillar myopathy and dilated cardiomyopathy [8,12]. Patients with heterozygous p.Pro209Leu mutations show the most severe clinical symptoms, such as progressive skeletal muscle weakness, cardiomyopathy, and severe respiratory insufficiency [8,12]. Diverse types of mutations have been reported in patients with dilated cardiomyopathy, including missense, truncation, and frameshift mutations, most of which are present in the BAG domain of BIS [8,12]. On a molecular basis, BIS colocalizes with z-disk proteins such as α-actinin and desmin [10]. Moreover, BIS interacts with HSC70, HSPB8, and CHIP to form the chaperone-assisted autophagy (CASA) complex, which is involved in the degradation of damaged filamin C [13]. BIS also contributes to maintaining myofibrillar integrity by interacting with the actin-capping protein, CapZ, under mechanical stress [14]. Moreover, BIS mediates myoblast mechanotransduction by regulating YAP and TAZ localization [15]. In addition to z-disk proteins, BIS was found to bind with β1-adrenergic receptors, L-type Ca^2+^ channels, and phospholemman, thereby modulating myocyte contraction [16]. These results show that BIS is essential for preserving the structural integrity of sarcomeres as well as the contractile function of myofibrils.

Recently, heart- or liver-specific *Bis*-KO mice were generated by us and other laboratories [17,18,19], demonstrating the impairment of autophagy and alteration of protein homeostasis as a common phenotype. However, attempts to generate skeletal muscle-specific *Bis*-KO mice have not been reported. Therefore, in this study, we generated skeletal muscle-specific *Bis*-KO mice using MLC-Cre mice. We found that depletion of BIS restricted to skeletal muscles led to early lethality, probably owing to respiratory failure accompanied by growth retardation, kyphosis, and a lack of peripheral fat-reproducing key phenotypes in whole-body *Bis*-KO mice. Mechanistically, the impairment of autophagy and metabolism, and accumulation of heat shock proteins and z-disk proteins were observed in the diaphragm and quadriceps. Our findings emphasize the importance of protein homeostasis regulated by BIS in maintaining proper skeletal muscle function.

## 2. Results

### 2.1. Bis-SMKO Mice Exhibited Growth Retardation and Early Lethality

By mating myosin light chain (MLC)-Cre mice with *Bis* exon 4-floxed mice, we generated skeletal muscle-specific *Bis*-KO (*Bis*-SMKO) mice, where BIS depletion was restricted to the skeletal muscles. Western blotting showed that BIS expression was significantly decreased in the diaphragm and quadriceps of *Bis*-SMKO mice, but its levels were similar in the liver, lung, and heart in control and *Bis*-SMKO mice (Figure 1A).

The incomplete BIS deletion in the diaphragm and quadriceps may be due to the remaining BIS proteins in slow-twitch muscle fibers because MLC-Cre activity is restricted to fast-twitch muscle fibers [20]. Double-labeling immunohistochemistry for BIS and slow myosin or fast myosin in the diaphragm indicates that BIS expression was diminished specifically in fast-twitch muscle fibers while it was retained in slow-twitch muscle fibers (Figure 1A). The body weights of *Bis*-SMKO mice were comparable to those of control mice at birth; however, growth retardation in *Bis*-SMKO mice became evident 26 days after birth (Figure 1B). The lifespan of *Bis*-SMKO mice was variable; the death of *Bis*-SMKO mice was observed as early as 30 days after birth, and all mice died within 4 months. Specifically, kyphosis and diminished peripheral fat were observed in *Bis*-SMKO mice compared to control mice (Figure 1C). *Bis*-SMKO mice exhibited dyspnea and abrupt weight loss before death. Therefore, *Bis*-SMKO mice reproduced the key phenotypes of whole-body *Bis*-KO mice, including growth retardation, kyphosis, diminished peripheral fat, respiratory failure, and early lethality [10,11].

Thirty-five days after birth, the mean body weight of the surviving *Bis*-SMKO mice was 14 g, which was 79% of that in control mice (Figure 1D). Moreover, 72% (N = 18 out of a total of 25) and 46% (N = 11 out of a total of 24) of the surviving male and female mice, respectively, showed weight loss at this point compared to their peak weights. Therefore, male mice were more susceptible to BIS depletion than female mice. We measured the weights of several organs, including the heart, liver, lung, thymus, diaphragm, and quadriceps from *Bis*-SMKO mice, at 35 days; the weights of each organ relative to the body weight were not significantly different from those of the control mice (Figure 1D). Notably, the relative kidney weight was higher in *Bis*-SMKO mice than in control mice.

Despite the loss of peripheral fat, the nutritional status of *Bis*-SMKO mice appeared normal, as shown by the blood levels of several metabolites, including total protein, albumin, glucose, total cholesterol, and triglycerides (Figure 1E). Blood chemistry also showed that liver function was not affected by BIS depletion in the skeletal muscles, as determined by the activities of aspartate aminotransferase, alanine aminotransferase, and gamma-glutamyl transpeptidase. The mean creatine kinase (CK) activity in *Bis*-SMKO mice did not exceed that in control mice; however, among the ten mice examined, two exhibited significantly higher levels of CK activity, indicating that at 35 days, some mice experienced severe muscle damage. Unexpectedly, serum lactate dehydrogenase (LDH) activity in *Bis*-SKMO mice was significantly decreased to 0.3-fold compared to that in control mice. In addition, the levels of blood urea nitrogen and creatinine in *Bis*-SMKO mice increased 1.4-fold and 1.9-fold, respectively, indicating abnormal kidney function. However, considering that the levels of total protein and albumin in *Bis*-SMKO mice were slightly elevated compared to those in the control, it cannot be excluded that *Bis*-SKMO mice were in dehydrated condition at 35 days after birth.

### 2.2. Muscle Degeneration and Regeneration Were Observed in Bis-SMKO Skeletal Muscles

Next, we performed histological analysis of the diaphragm and quadriceps of control and *Bis*-SMKO mice. Hematoxylin and eosin (H & E) staining revealed that muscle fibers with small diameters and centralized nuclei were frequently observed in the diaphragm and quadriceps of *Bis*-SMKO mice (Figure 2A,D). Specifically, the mean cross-sectional area (CSA) decreased from 509.2 μm^2^ to 309.5 μm^2^ in the diaphragm and from 748.1 μm^2^ to 614.2 μm^2^ in the quadriceps of *Bis*-SMKO mice compared to control mice (Figure 2B,E). The histogram of fiber diameter distribution showed that fibers with smaller diameters were increased in *Bis*-SMKO mice in both the diaphragm and quadriceps but more evidently in the diaphragm (Figure 2B,E). The percentage of fibers with centralized nuclei increased from 0.21% to 7.6% in the diaphragm and from 0.25% to 5.2% in the quadriceps of *Bis*-SMKO mice compared to the control (Figure 2B,E), indicating that muscle regeneration occurred in *Bis*-SKMO mice.

We also examined regeneration markers using quantitative real-time reverse transcription polymerase chain reaction (qRT-PCR). The mRNA expression of embryonic myosin heavy chain (MYH3) in the diaphragm and quadriceps of *Bis*-SMKO mice increased 251-fold and 46-fold, respectively, compared to that in control mice (Figure 2C,F). The mRNA expression of myogenin was significantly elevated in the diaphragm but to a lesser degree in the quadriceps of *Bis*-SMKO mice. Therefore, muscle regeneration occurred in both the diaphragm and quadriceps but more prominently in the diaphragm of *Bis*-SMKO mice.

Muscle regeneration occurs in response to injured stimuli [21]. To identify muscle degeneration, we first examined the levels of cleaved PARP1, an apoptosis marker, by immunohistochemistry. *Bis*-SMKO diaphragm exhibited a notable increase in immunostaining intensities for cleaved PARP1, indicating damaged muscle fibers (Figure 2G). An increase in cleaved PARP1 levels was also demonstrated by Western blotting (Figure 2H). Therefore, the *Bis*-SMKO diaphragm underwent muscle degeneration and consequent muscle regeneration.

### 2.3. Accumulation of Heat Shock Proteins (HSPs) and z-Disk Proteins in Bis-SMKO Skeletal Muscles

To investigate the degenerating nature of the myofibrils from *Bis*-SMKO mice, we performed H & E staining on longitudinal sections of the diaphragm. Bulb-like protrusions with disrupted myofibril integrity were frequently observed within the fascicles of *Bis*-SMKO skeletal muscles (Figure 3(A-4)). To further examine myofibrillar disruption, transmission electron microscopy (TEM) was performed. Ultrastructural myofibril organization, including Z-lines, in *Bis*-SMKO mice was completely lost compared to that in control mice (Figure 3(A-2,A-5,A-7)).

Interestingly, degenerated and swollen mitochondria, which exhibited disruption of crista and decreased electron density, suggesting the loss of matrix proteins, were frequently identified within the degenerated myofibrils in the diaphragm of *Bis*-SMKO mice (Figure 3(A-6)). Western blotting showed that the expression of mitochondrial marker protein COX4 was decreased in the diaphragm of *Bis*-SMKO mice (Appendix A), indicating that part of mitochondrial proteins was lost in *Bis*-SMKO mice. In addition, double-membrane autophagic vacuoles were observed in the degenerated myofibrils of *Bis*-SMKO mice (Figure 3(A-8,A-9)). Enlarged mitochondria have also been previously described in the liver of hepatocyte-specific *Atg5*-KO mice [22], indicating that abnormalities in mitochondria structures may be associated with autophagy impairment in *Bis*-SKMO mice.

In the hearts and livers of heart- and liver-specific *Bis*-KO mice, the expression of p62 increased, while that of HSPB8 decreased [17,19]. Furthermore, p62 protein expression increased in the diaphragm and quadriceps of *Bis*-SMKO mice (Figure 3B and Appendix A). However, the p62 mRNA levels did not change significantly. This indicates that the accumulation of p62 may be caused by autophagy impairment but not by a transcriptional mechanism. HSPB8 protein expression, which is stabilized by its interaction with BIS [17,23], decreased, but HSPB5 and HSP70 protein expression increased in the diaphragm and quadriceps (Figure 3B and Appendix A). The mRNA levels of HSPB8, HSPB5, and HSP70 increased more than 2.5-fold in the diaphragm of *Bis*-SMKO mice compared to that in control mice (Figure 3C). The quadriceps of *Bis*-SMKO mice also exhibited elevated mRNA levels in the HSPs (Appendix A).

Filamin C, located in the z-disks, is a client protein for CASA, where BIS is involved [13]. Western blotting showed that filamin C accumulated in the diaphragm and quadriceps of *Bis*-SMKO mice (Figure 3B and Appendix A). The accumulation of another z-disk protein, desmin, was more evident in the diaphragm than in the quadriceps of *Bis*-SMKO mice (Figure 3B and Appendix A).

Collectively, autophagy was impaired, and heat shock proteins, except HSPB8 and z-disk proteins, accumulated in *Bis*-SMKO skeletal muscles.

### 2.4. YAP1 Signaling Was Enhanced in Bis-SMKO Diaphragm

To elucidate the underlying mechanism of HSPB5 and HSP70 accumulation, we focused on the increase in mRNA expression of HSPB8, HSPB5, and HSP70 in *Bis*-SMKO mice. HSF1 and YAP1 are transcription factors that regulate heat shock protein transcriptome [24]. To address whether HSF1 and YAP1 were activated in *Bis*-SMKO mice, we first carried out Western blotting and observed an obvious increase in YAP1 expression in the diaphragm of *Bis*-SMKO mice compared with that in control (Figure 4A). HSF1 expression did not notably change in the diaphragm of *Bis*-SMKO mice (Figure 4A).

We confirmed YAP1 expression by immunohistochemistry and observed an overall increase in YAP1 immunostaining intensity in the *Bis*-SMKO diaphragm (Figure 4B). To identify the functionality of increased YAP1 expression, we examined the mRNA expression of well-known YAP1 target genes. Compared to the control, the mRNA expression of CTGF, CYR61, and ANKRD1 was significantly elevated in the diaphragm in *Bis*-SMKO mice (Figure 4C). In the quadriceps, the expression of YAP1 was not notably altered in *Bis*-SMKO mice (Appendix A). However, the downstream targets of YAP1 were significantly increased by BIS depletion (Appendix A). Therefore, in the skeletal muscles of *Bis*-SMKO mice, activation of YAP1 signaling may be involved in the induction of HSPs.

### 2.5. Bis-SMKO Skeletal Muscles Exhibited Defective Energy Metabolism

A previous study has reported that *Hspb8*-knockout hearts have impaired energy metabolism [25]. Furthermore, BIS directly regulates metabolism by stabilizing hexokinase 2 mRNA and glutaminase in cancer cells [26,27]. Considering that *Bis*-SMKO skeletal muscles have diminished HSPB8 levels, we hypothesized that BIS depletion might directly or indirectly affect energy metabolism.

First, we found that ATP levels in the diaphragm and quadriceps of *Bis*-SMKO mice were decreased to 53% and 42% of those in control mice, respectively. However, ATP levels in the heart of *Bis*-SMKO mice did not differ from those of control mice (Figure 5A). Therefore, ATP levels were specifically reduced in skeletal muscles depending on the BIS depletion.

As shown in Figure 1E, LDH activity in the serum of *Bis*-SMKO mice was significantly lower than that in control mice. We also measured LDH and CK activities in the diaphragm and quadriceps of control and *Bis*-SMKO mice. The LDH activity in the diaphragm and quadriceps significantly decreased to 17% and 27% of that in control mice, respectively (Figure 5B,C). CK activity in *Bis*-SMKO mice significantly decreased to 40% of control (13,200 U/g difference) in the diaphragm but only decreased to 79% of control (16,135 U/g difference) in the quadriceps (Figure 5B,C). The basal CK activity in the quadriceps was higher than that in the diaphragm, which might temper the differences in CK activity between the two groups in the quadriceps. Therefore, *Bis*-SMKO skeletal muscles had energy metabolism defects with decreased ATP levels and LDH and CK activities.

## 3. Discussion

In the present study, we generated skeletal muscle-specific *Bis*-deficient mice that reproduced the main phenotypes of whole-body *Bis*-KO mice, including growth retardation, kyphosis, diminished peripheral fat, respiratory failure, and early lethality. Therefore, the early death of whole-body *Bis*-KO and *Bis*-SMKO mice can be attributed to the depletion of BIS and subsequent dysfunction of skeletal muscles. While whole-body *Bis*-KO mice exhibit severe metabolic deterioration after birth [11], the general nutritional status of *Bis*-SMKO mice was within the normal range in terms of serum levels of glucose, total proteins, and cholesterol at 35 days after birth. Differences in metabolic status may be due to the sucking power exerted by slow-twitch fibers where BIS expression is preserved. Another explanation is that water and food were supplementary provided at the bottom of the cages to increase feeding accessibility. However, growth retardation and a decrease in peripheral fat indicate that food intake in *Bis*-SMKO mice seems insufficient for normal growth.

Skeletal muscle dysfunction in *Bis*-SMKO mice has been demonstrated in several observations. First, respiratory insufficiency was observed in most *Bis*-SMKO mice before death. Second, apoptosis was increased in the diaphragm of *Bis*-SKMO mice, and the expression of regeneration markers such as MYH3 and myogenin was induced, accompanied by an increase in the regenerating myofibers with centralized nuclei. Third, electron microscopy revealed the disruption of sarcomere structures with the loss of z-disks in the myofibrils, indicating an attenuated contractile force. The skeletal muscles of whole-body *Bis*-KO mice generated in our previous study also exhibited disturbances in z-disk alignment but rare muscle regeneration and slight apoptosis [11]. One possible explanation for this discrepancy is the observation points. The histology of skeletal muscle was examined in whole-body *Bis*-KO mice at 14 days when skeletal muscle degeneration did not progress significantly. Considering that BIS depletion is restricted to fast-twitch muscle fibers in *Bis*-SMKO mice, it cannot also be excluded that BIS in satellite cells has the potential to drive muscle regeneration processes, including satellite cell proliferation and differentiation into myoblasts and, ultimately, fiber regeneration. In support of this possibility, we previously reported that BIS depletion in hepatocytes induces senescence, suggesting that BIS plays a role in maintaining liver regeneration potential [19]. Therefore, the role of BIS in skeletal muscle regeneration requires further investigation.

Disturbances in protein homeostasis have been previously suggested as a common critical finding in liver- or heart-specific *Bis*-KO mice [17,18,19]. Autophagy impairment was also observed in *Bis*-SMKO skeletal muscles, as shown by the accumulation of p62 and frequent observation of autophagic vacuoles by electron microscopy. In addition, alterations in the expression of HSPs were observed in *Bis*-SMKO mice; Western blotting revealed that HSPB8 expression was reduced, whereas HSPB5 and HSP70 expression was increased. The decrease in HSPB8, which was stabilized by BIS, was consistent with the results in heart- or liver-specific *Bis*-KO mice [17,18,19]. In heart-specific *Bis*-KO mice, HSPB5 and HSP70 levels increased specifically in the insoluble fraction [17]. Therefore, the increase in the total amount of these proteins in *Bis*-SMKO mice likely included the aggregated forms of both HSPs. The results of the current and previous studies show that BIS is essential for the stabilization of HSPB8 and the preservation of HSPB5 and HSP70 solubility. Based on the fact that filamin C and desmin were also found in the insoluble fraction of heart-specific *Bis*-KO mice [17,18], the accumulation of filamin C and desmin in *Bis*-SMKO mice may be aggregated in non-functional forms, probably owing to autophagy impairment and insufficient chaperone proteins, thereby affecting the contractile forces of skeletal muscles.

Interestingly, the mRNA expression of heat shock proteins, including HSPB8, HSPB5, and HSP70, was increased in the diaphragm and quadriceps of *Bis*-SMKO mice. Therefore, the accumulation of HSPB5 and HSP70 appears to be caused by increased mRNA expression as well as protein aggregation. The expression profiles of HSPB5 and HSP70 were similar to those of transgenic mice overexpressing human BIS (P209L), the substitution of which has been reported in some patients with fulminant myopathy [8,12]. Overexpression of BIS (P209L) in the hearts and skeletal muscles leads to growth retardation, restrictive cardiomyopathy, and early lethality. In addition, hearts with overexpressed BIS (P209L) exhibited sarcomere disruption and accumulation and aggregation of proteins, including HSPB5 and HSP70, accompanied by an increase in their mRNA expression [28]. Therefore, proline at p.209 in BIS is critical for protein homeostasis, and its substitution or loss may lead to proteotoxic stress. On this basis, the transcriptional activation of heat shock proteins may be a common compensatory mechanism against proteotoxic stress.

Although HSF1 is the main transcriptional activator of heat shock proteins, recent findings indicate that YAP1 is also involved in inducing the heat shock transcriptome [24]. Here, the expression of YAP1, rather than that of HSF1, was increased in the diaphragms of *Bis*-SMKO mice, as determined by Western blotting and immunostaining. Considering that several downstream targets of YAP1 were also increased in *Bis*-SMKO mice, the activation of YAP1 seems to be responsible, at least in part, for the increase in HSPB8, HSPB5, and HSP70 mRNA levels in *Bis*-SMKO mice. BIS depletion has been found to reduce the nuclear translocation of YAP1 [29]. Therefore, the absence of BIS is unlikely to directly promote the nuclear translocation of YAP1 in our in vivo model. Increased YAP1 expression, nuclear localization, and enhanced YAP1 downstream signaling have also been found in dystrophic skeletal muscles [30]. Furthermore, YAP1 expression increased upon denervation to protect against muscle atrophy [31]. Therefore, YAP1 activation may be a common response to abnormal muscle physiology, of which molecular details remain to be determined.

Another significant finding of our study is that BIS depletion lowers ATP levels and LDH and CK activities in skeletal muscles. Although BIS directly stabilizes hexokinase 2 mRNA and glutaminase in cancer cells [26,27], the molecular mechanism of defective energy metabolism under BIS depletion is unclear. The indirect regulation of metabolism by BIS via HSPB8 may be a possible mechanism because HSPB8 expression notably decreased in *Bis*-SMKO skeletal muscles. In addition, our model and *Hspb8*-KO mice shared similar metabolism-related defects. Both our model and *Hspb8*-KO hearts exhibited decreased ATP levels [25]. Furthermore, several genes involved in glycolysis and fatty acid metabolic pathways that were affected in *Hspb8*-KO hearts were altered in *Bis*-SMKO skeletal muscles. In addition, both our model and *Hspb8*-KO skeletal muscles showed enlarged and degenerated mitochondria [32]. Therefore, the loss of HSPB8 following BIS depletion may be a crucial factor in metabolic impairment. Nevertheless, direct regulation of metabolic genes by BIS is plausible. Based on the BioGRID database (thebiogrid.org, accessed on 29 May 2023), BIS and lactate dehydrogenase A (LDHA) have numerous overlapping interacting partners, including BAG1, FZR1, HSP70, HSP90, p62, PKM, PARK2, RAB1A, TRIM67, YAP1, and YWHAQ. BIS and creatine kinase, M-type (CKM), also have common binding partners such as FZR1, HSPB2, LATS1, and YWHAQ. Therefore, through the complicated networks between BIS and LDH or CK, BIS could regulate the protein stabilities of these two proteins. However, the exact molecular mechanisms are in the scope of further studies.

In conclusion, for the first time, we generated skeletal muscle-specific *Bis*-KO mice. This mouse model revealed that BIS in skeletal muscle is critical for survival and reinforces the important role of BIS in protein homeostasis. We hope that this model will contribute to the development of therapeutic strategies for myopathy and reveal novel functions of BIS in skeletal muscle physiology.

## 4. Materials and Methods

### 4.1. Animals

Mice homozygous for exon 4-floxed *Bis* (*Bis*^f/f^) were crossbred with MLC-Cre transgenic mice to obtain *Bis*^f/+^; MLC-Cre mice, which were subsequently interbred to produce skeletal muscle-specific *Bis*-knockout mice (*Bis*^f/f^; MLC-Cre, *Bis*-SMKO). Male *Bis*-SMKO mice aged 35 days and age-matched *Bis*^f/f^ mice (control) were used in this study unless otherwise stated. To minimize the effect of nutritional status on the skeletal muscles, supplementary food was applied to the bottom of the cage after 21 days, and weaning was not performed until 35 days. All procedures and provisions for animal care were approved by the Institutional Animal Care and Use Committee of the College of Medicine of the Catholic University of Korea (CUMS-2017-0320-10) and conformed to the ARRIVE and the National Institutes of Health guidelines (NIH Publications No. 8023, revised 1978).

### 4.2. Biochemical Analysis

To determine the levels of serum metabolites and several biochemical markers, blood samples were collected from the inferior vena cava of control and *Bis*-SMKO mice and analyzed by DK Korea (Seoul, Korea) using an AU480 chemistry analyzer (Beckman Coulter, Brea, CA, USA). Lactate dehydrogenase and creatine kinase activities were measured in the diaphragm and quadriceps of control and *Bis*-SMKO mice. ATP levels were measured using the PicoSensTM ATP Assay Kit (BIOMAX, Guri-si, Korea) according to the manufacturer’s protocol in the diaphragm, quadriceps, and heart of the control and *Bis*-SMKO mice.

### 4.3. Electron Microscopy

Electron microscopy was performed as described in a previous study [11]. Briefly, the diaphragm sections embedded in Epon 812 (Polysciences, Warrington, PA, USA) were observed and recorded using a TEM (JEM 1010; JEOL Ltd., Tokyo, Japan) and a CCD camera (SC1000; Gatan, Pleasanton, CA, USA).

### 4.4. Histological Analysis

Hematoxylin and eosin (H & E) staining was performed on paraffin-embedded sections of the diaphragm and quadriceps. Immunohistochemistry of paraffin-embedded diaphragm sections was performed using antibodies against BIS [1], slow myosin, fast myosin (Sigma-Aldrich, St. Louis, MO, USA), cleaved PARP1 (Abcam, Cambridge, UK), HSF1 (Enzo Life Sciences, Farmingdale, NY, USA), and YAP1 (Santa Cruz Biotechnology, Dallas, TX, USA). Cell nuclei were counterstained with DAPI (4′,6-diamidino-2′phenylindole, 1:2000; Roche, Mannheim, Germany) for 10 min in immunofluorescence immunohistochemistry. Bright-field images were captured using Pannoramic MIDI (3DHISTECH Ltd., Budapest, Hungary) and analyzed using CaseViewer 2.4 software (3DHISTECH Ltd.). Fluorescence images were viewed under a confocal microscope (LSM 900 with Airyscan; Carl Zeiss Co. Ltd., Oberkochen, Germany) equipped with four lasers (Diode 405, Argon 488, HeNe 543, and HENe 633) under constant viewing conditions. The CSA, minimum Feret diameter, and percentage of fibers with centralized nuclei were measured from multiple areas using Image J software (Fiji; National Institutes of Health, Bethesda, MD, USA).

### 4.5. Quantitative Real-Time Reverse Transcription Polymerase Chain Reaction (qRT-PCR)

To analyze mRNA expression, skeletal muscle tissues were homogenized with RNAiso Plus (Takara Bio, Kusatsu, Japan), and mRNAs were extracted according to the manufacturer’s protocol. cDNAs were synthesized using PrimeScript RT Master Mix (Takara Bio) according to the manufacturer’s protocol. qRT-PCR was performed using the synthesized cDNAs, gene-specific primers, and TB Green Premix Ex Taq (Takara Bio) using the CFX Connect Real-Time System (Bio-Rad, Hercules, CA, USA). Gene-specific primer sequences are listed in Appendix A.

### 4.6. Western Blotting

For Western blotting, skeletal muscle tissues were homogenized with RIPA buffer. After the lysates were centrifuged at 13,000 rpm for 30 min, the supernatants were measured using bicinchoninic acid (BCA) (Thermo Fisher Scientific, Waltham, MA, USA) assay and boiled for 5 min at 100 °C after mixing with a 5× sample buffer. Western blotting samples were loaded onto SDS-PAGE gels, run, and transferred onto PVDF membranes (Merck Millipore, Burlington, MA, USA). After blocking with Tris Buffered Saline with Tween 20 (TBST) containing 5% skim milk, the membranes were incubated with primary antibodies overnight at 4 °C and then incubated with secondary antibodies for 1–2 h at RT. The membranes were incubated with ECL solution (Promega, Madison, WI, USA) and imaged using ImageQuant LAS 500 (GE Healthcare, Chicago, IL, USA).

The primary antibodies used were anti-BIS [1], anti-HSF1, anti-HSP70 (Enzo Life Sciences), anti-GAPDH, anti-HSPB5, anti-YAP1 (Santa Cruz Biotechnology), anti-cleaved PARP1, anti-desmin, anti-HSPB8, anti-p62 (Abcam), anti-COX4 (Cell Signaling Technology, Danvers, MA, USA), and anti-filamin C (Novus Biologicals, Centennial, CO, USA).

### 4.7. Statistical Analysis

Data are presented as mean ± SEM (standard error of the mean). A two-tailed Student’s *t*-test was used to compare two different groups. Statistical significance was set at *p* < 0.05.

## Figures and Tables

**Figure 1 ijms-24-09635-f001:**
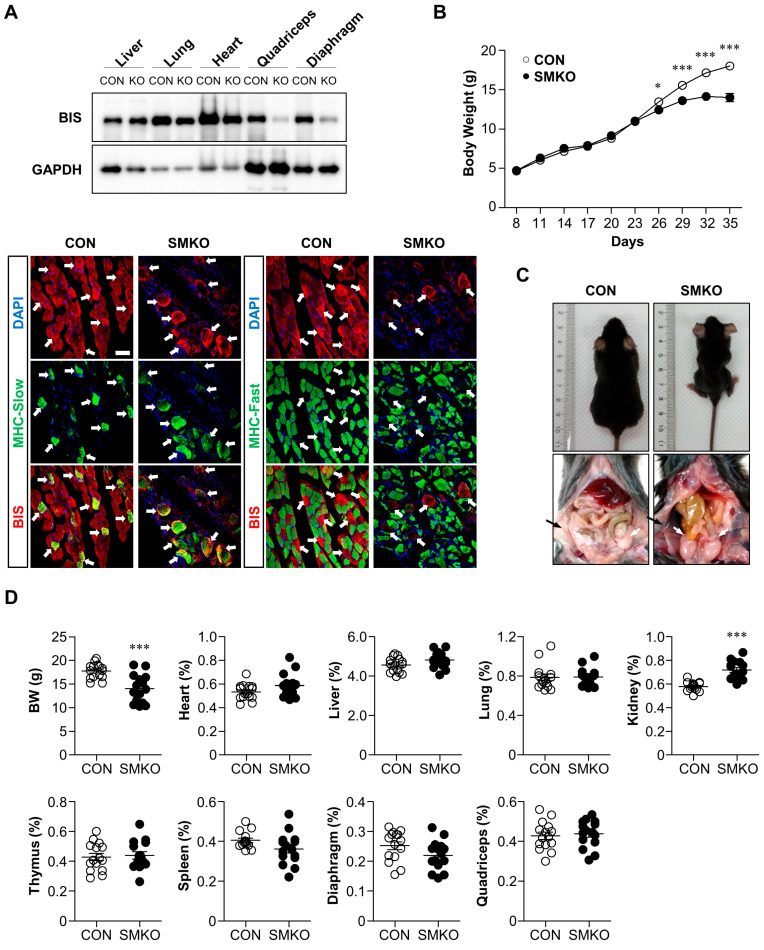
General phenotypes of *Bis*-SMKO mice. (**A**) Skeletal muscle-specific BIS depletion in *Bis*-SMKO mice was analyzed through Western blotting (**top**). Liver, lung, heart, quadriceps, and diaphragm from male control and *Bis*-SMKO mice at 29 days were analyzed via Western blotting. GAPDH, glyceraldehyde 3-phosphate dehydrogenase. Double-labeling immunohistochemistry for BIS and slow myosin heavy chain (MHC-Slow) or fast myosin heavy chain (MHC-Fast) in the diaphragm shows BIS expression was retained in slow-twitch muscle fibers while BIS expression was lost in fast-twitch muscle fibers in *Bis*-SMKO mice (bottom). White arrows indicate slow-twitch muscle fibers. Note disorganized MHC-fast immunoreactivity with loss of BIS expression in *Bis*-SMKO mice. Scale bars, 40 μm. (**B**) Growth curve of control and *Bis*-SMKO mice. Mice were weighed at 3-day intervals from 8 days after birth to 35 days (N = 26 for both groups, 13 males and 13 females in each group). (**C**) Representative pictures of male control and *Bis*-SMKO mice at 33 days. *Bis*-SMKO mice had relatively small body sizes and kyphosis (**top**), and diminished peripheral fat (**bottom**). The black arrow indicates inguinal fat, and the white arrows indicate epididymal fat. (**D**) The body weights and relative weights of the heart, liver, lung, kidney, thymus, spleen, diaphragm, and quadriceps were measured. The weights of each organ were divided by body weights and presented as % (N = 15 for both groups: 8 males and 7 females in each group). BW, body weights. (**E**) Blood biochemistry was performed to determine the levels of total protein, albumin, glucose, total cholesterol (cholesterol), triglycerides, aspartate aminotransferase (AST), alanine aminotransferase (ALT), gamma-glutamyl transpeptidase (GGT), creatine kinase (CK), lactate dehydrogenase (LDH), blood urea nitrogen (BUN), and creatinine in control and *Bis*-SMKO mice (N = 10 for both groups, 5 males and 5 females in each group). CON, control; SMKO, *Bis*-SMKO. Data are presented as mean ± SEM (standard error of the mean). * *p* < 0.05, ** *p* < 0.01, and *** *p* < 0.001 compared to the control.

**Figure 2 ijms-24-09635-f002:**
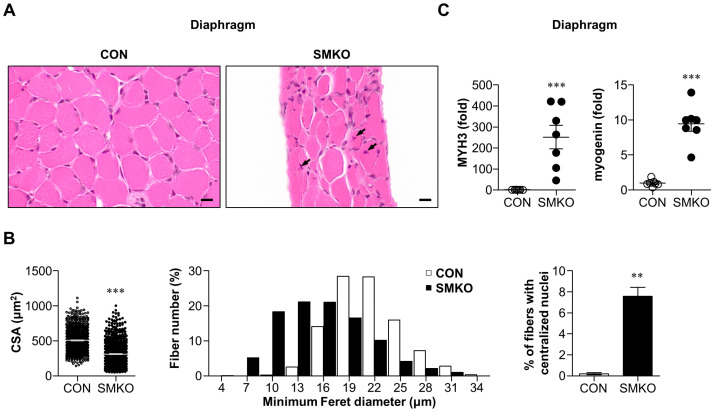
Depletion of BIS in skeletal muscles resulted in muscle regeneration and apoptosis. H & E staining showed regenerating fibers having centralized nuclei in the *Bis*-SMKO diaphragm (**A**) and quadriceps (**D**) compared to the control. Black arrows indicate regenerating fibers with centralized nuclei. Scale bars, 10 μm. The cross-sectional area (CSA; **left**), minimum Feret diameter (**middle**), and % of fibers with centralized nuclei (**right**) were measured in more than 750, 750, and 1500 fibers, respectively (N = 3 for both groups) in the diaphragm (**B**) and quadriceps (**E**) of control and *Bis*-SMKO mice. The mRNA expression of MYH3 and myogenin was analyzed through quantitative real-time reverse transcription polymerase chain reaction (qRT-PCR) in the diaphragm (**C**) and quadriceps (**F**) of control and *Bis*-SMKO mice (N = 7 for both groups). Data were normalized relative to GAPDH mRNA expression, and the corresponding values of control mice were designated as 1.0. (**G**) The representative result of immunostaining with antibody for cleaved PARP1 in the diaphragm of *Bis*-SMKO and control mice. Scale bars, 10 μm. (**H**) Western blotting for cleaved PARP1 levels in the diaphragm of *Bis*-SMKO and control mice. cPARP1, cleaved PARP1. CON, control; SMKO, *Bis*-SMKO. Data are presented as mean ± SEM. * *p* < 0.05, ** *p* < 0.01, and *** *p* < 0.001 compared to control.

**Figure 3 ijms-24-09635-f003:**
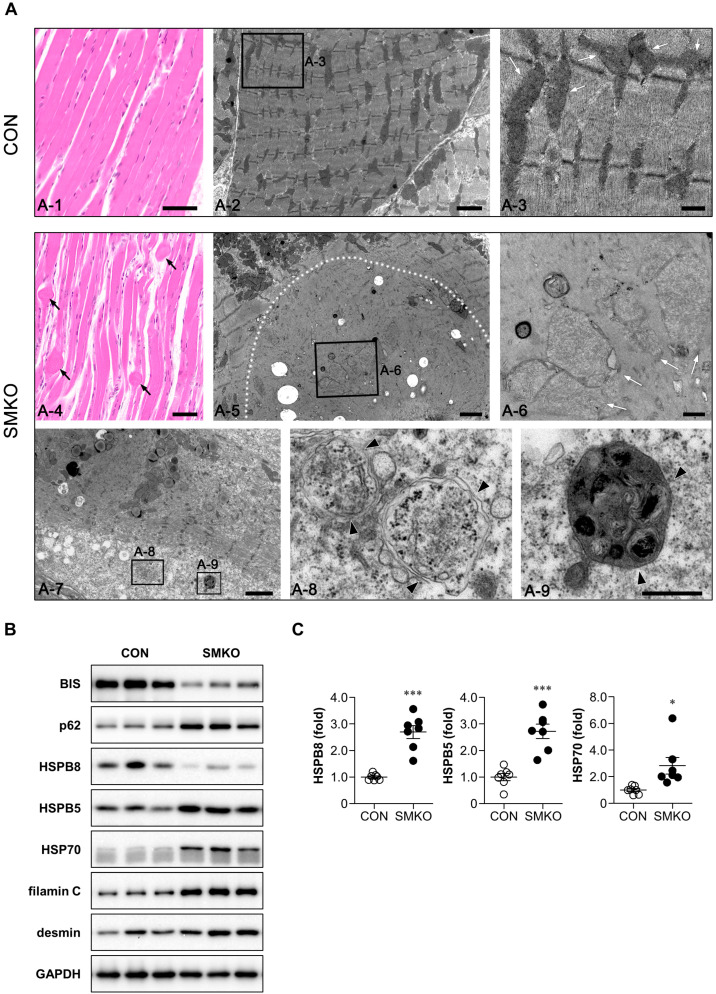
Protein quality control was impaired in *Bis*-SMKO mice. (**A**) Representative images of H & E and TEM analysis of diaphragm of control and *Bis*-SMKO mice at 35 days: (**A-1**,**A-4**) Longitudinal H & E section shows bulb-like protrusions (black arrows in **A-4**) with disrupted myofibril integrity in *Bis*-SMKO mice. (**A-2**,**A-3**) TEM analysis of the control mice shows organized Z-lines and electron-dense mitochondria with intact crista (white arrows in **A-3**). (**A-5**,**A-6**) TEM analysis of *Bis*-SMKO mice shows disorganized myofibrils with the loss of Z-lines. A White dotted line marks a bulb-like protrusion of degenerated myofibril. White arrows in (**A-6**) show swollen mitochondria with disrupted crista and decreased electron density. (**A-7**,**A-8**,**A-9**) Double-membrane autophagosomes (black arrowheads in **A-8**,**A-9**) with different morphologies are frequently found within the degenerated myofibrils with complete loss of fibrillar structures (scale bars, 50 μm for **A-1**,**A-4**; 2 μm for **A-2**,**A-5**,**A-7**; 0.5 μm for **A-3**,**A-6**,**A-8**,**A-9**). (**B**) Expression of BIS, p62, HSPB8, HSPB5, HSP70, filamin C, desmin, and GAPDH in the diaphragm of control and *Bis*-SMKO mice was analyzed by Western blotting. (**C**) mRNA expression of HSPB8, HSPB5, and HSP70 in the diaphragm of control and *Bis*-SMKO mice was analyzed by qRT-PCR (N = 7 for both groups), as shown in Figure 2C. CON, control; SMKO, *Bis*-SMKO. Data are presented as mean ± SEM. * *p* < 0.05 and *** *p* < 0.001 vs. control.

**Figure 4 ijms-24-09635-f004:**
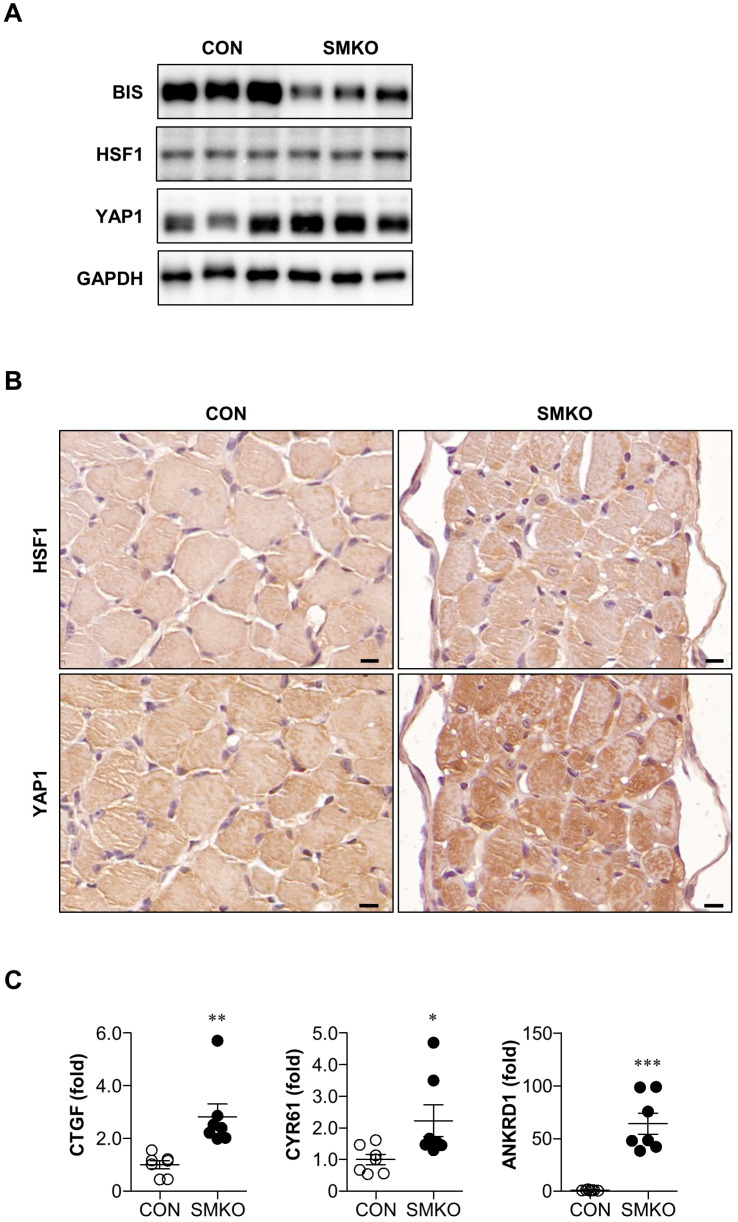
YAP1 signaling was activated in the *Bis*-SMKO diaphragm. (**A**) Expression of BIS, HSF1, YAP1, and GAPDH in the diaphragm of control and *Bis*-SMKO mice was accessed by Western blotting. (**B**) HSF1 and YAP1 expression was examined by immunohistochemistry using serial paraffin sections in the diaphragm of control and *Bis*-SMKO mice. Scale bars, 10 μm. (**C**) mRNA expression of YAP1 downstream targets, including CTGF, CYR61, and ANKRD1 in the diaphragm of control and *Bis*-SMKO mice, was analyzed by qRT-PCR (N = 7 for both groups), as shown in Figure 2C. CON, control; SMKO, *Bis*-SMKO. Data are presented as mean ± SEM. * *p* < 0.05, ** *p* < 0.01, and *** *p* < 0.001 compared to control.

**Figure 5 ijms-24-09635-f005:**
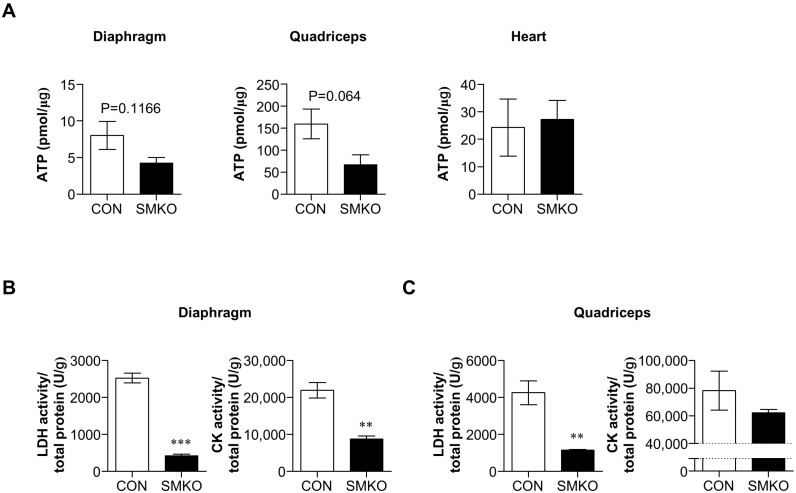
*Bis*-SMKO skeletal muscles exhibited metabolic impairment. (**A**) ATP levels were measured in the diaphragm, quadriceps, and heart of control and *Bis*-SMKO mice (N = 4 for all groups). Data were presented as relative amounts of ATP per microgram of protein. LDH and CK activities were measured from the diaphragm (**B**) and quadriceps (**C**) of control and *Bis*-SMKO mice (N = 4 for both groups). Data were expressed as relative activities per gram of protein. CON, control; SMKO, *Bis*-SMKO. Data are presented as mean ± SEM. ** *p* < 0.01 and *** *p* < 0.001 compared to control.

## Data Availability

The data presented in this study are available upon reasonable request from the corresponding author.

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
