# Peer review of "Skeletal Muscle-Specific Bis Depletion Leads to Muscle Dysfunction and Early Death Accompanied by Impairment in Protein Quality Control"

_ijms, 2023, doi:10.3390/ijms24119635_

Round 1

Reviewer 1 Report

In the article “Skeletal muscle-specific Bis depletion leads to muscle dysfunction and early death accompanied by impairment in protein quality control” Jung and colleagues created for the first time a mice lacking BIS specifically in the skeletal muscle and demonstrated that Bis plays a crucial role in the development of the skeletal muscle. The article is well written, however, in my opinion, before the publication in International Journal of Molecular Sciences, the authors should include some analyses.

·         I suggest to perform the count of the centrally-nucleated myofibers in the diaphragm and quadriceps to asses that in the Bis-SKMO mice there is an increase of the regenerating myofibers, if is necessary should also be divided between the big and small centrally-nucleated myofibers representing the regenerated and regenerating, respectively;

·         even if there is no difference in the muscle weight, should be interesting to analyse the cross sectional area (CSA) of the myofibers in the diaphragm and quadriceps, to observe if there is a reduction of the muscle mass in term of fiber dimension;

·         moreover, I suppose that is not easy to perform at the moment, but another important evaluation should be the muscle performance of the Bis-SKMO, to demonstrate that the absence of Bis reduce the skeletal muscle mass and performance.

Reviewer 2 Report

The manuscript describes the characteristics of a transgenic mouse with skeletal muscle-specific knockout of Bcl-2-interacting cell death suppressor (BIS), which clearly shows the potential as a model for the studies of the functions of BIS. This is a further development of the whole-body knockout mouse the authors previously described. The manuscripts merits publication, however, there are a few places need to be improved.

1. Line 115-116, the Bis-SMKO mice have a life span between 1 and 4 months, which is quite large range. Therefore it would be more informative if the actual survival curve could be presented here, along with the survival curve of the whole-body knockout.

2. Figure 1A, the Western blotting shows the clear reduction of BIS in skeletal muscles. As trace amount of BIS was still detected in the muscles, the authors suggest that it comes from slow-twitch muscle fibers. Did the authors perform the IHC on muscle sections to prove that was the case?

3. Line 122and 123, 72% of males and 46% of females of Bis-SMKO mice showed significant weight loss. What is the criteria to tell that a mouse has a significant weight loss, and what is the sample size?

4. Figure 2A, in addition to the H&E images, did the authors perform a statistic analysis showing the distribution of the diameter of muscle fibers and the percentage of the cells showing centralized nuclei?

5. Figure 3A, the mitochondria clearly show the change in morphology. Have the authors carried out molecular or functional studies of the mitochondria. For example, is there any change at protein levels of mitochondria markers?

Generally good, only minor changes are needed.

Round 2

Reviewer 1 Report

In the revised version of the article “Skeletal muscle-specific Bis depletion leads to muscle dysfunction and early death accompanied by impairment in protein quality control” Jung and colleagues performed most of the analysis requested improving the quality of the manuscript. Thus, the current version of the paper in my opinion is suitable for the publication in International Journal of Molecular Sciences.